# Evaluating the effect of Chinese control measures on COVID-19 via temporal reproduction number estimation

**Duanbing Chen[1,2], Tao Zhou [1,3]***

**1** Big Data Research Center, University of Electronic Science and Technology of China, Chengdu, Peoples' Republic of China, **2** Union Big Data, Chengdu, Peoples' Republic of China, **3** Tianfu Complexity Science Research Center, Chengdu, Peoples' Republic of China

* zhutou@ustc.edu

## Abstract

Control measures are necessary to contain the spread of serious infectious diseases such as COVID-19, especially in its early stage. We propose to use temporal reproduction number an extension of effective reproduction number, to evaluate the efficacy of control measures, and establish a Monte-Carlo method to estimate the temporal reproduction number without complete information about symptom onsets. The province-level analysis indicates that the effective reproduction numbers of the majority of provinces in mainland China got down to < 1 just by one week from the setting of control measures, and the temporal reproduction number of the week [15 Feb, 21 Feb] is only about 0.18. It is therefore likely that Chinese control measures on COVID-19 are effective and efficient, though more research needs to be performed.

## Introduction

Emerged from Wuhan City, the novel coronavirus diseases rapidly expanded since December 2019. Early analyses indicated that COVID-19 has middle-to-high transmissibility, with preliminary estimation of basic reproduction number $R_0$ lying in the range [2.0, 4.0], e.g., 1.4-3.9 [1], 2.47-2.86 [2] and 2.8-3.9 [3]. After a period of stealthy spread, on 20 January 2020, COVID-19 was identified as a B-type infectious disease in China, and the control measures were set according to the standard of A-type infectious disease. Roughly speaking, 21 January 2020 can be considered as the starting date of control, on which every province in China took COVID-19 spread as an emergency event and launched strong control measures according to directives of the central government. These control measures have achieved remarkable success, with daily number of confirmed cases quickly decreasing after a short expansion lasting about two weeks from 21 January 2020.

In general, basic reproduction number $R_0$ can be used to characterize the transmissibility of infectious diseases. It refers to the average number of individuals who will be infected by one infected case in a fully susceptible population without external interventions. Without control, infectious diseases will gradually die out if $R_0 < 1$, will spread exponentially and become

from email dbchen@uestc.edu.cn The authors of the present study had no special access privileges in accessing the datasets from the National Health Commission of China which other interested researchers would not have.

**Funding:** This work was partially supported by the National Natural Science Foundation of China (61673085, 11975071, 61433014) and by the Science Strength Promotion Programme of UESTC (Y03111023901014006). There was no additional external funding received for this study.

**Competing interests:** The authors have declared that no competing interests exist.

epidemics if $R_0 > 1$, and will become endemic in the population if $R_0 \approx 1$. The basic reproduction number is far different for different infectious diseases, for example, Zika: 1.4-6.6 [4], H1N1: 1.4-3.1 [5], dengue: 1.52-3.90 [6], Ebola: 1.3-2.7 [7], SARS: 2.2-3.7 [8], MERS: 2.0-6.7 [9], smallpox: 3.5-6.0 [10], measles: 12-18 [11], pertussis: 12-17 [12], etc. Usually, it is difficult to directly measure the value of $R_0$ since $R_0$ is affected by numerous biological, sociobehavioral, and environmental factors [13], and thus statistical models are widely applied to estimate $R_0$ [14–17].

We always assume the population is fully susceptible without control measures in estimating the value of $R_0$. However, during the epidemic spreading, various control measures will be introduced to contain the spread, so we should adopt time-related reproduction number to quantify the temporal situation of the spread and the control efficacy. The most intuitive metric is the effective reproduction number $R_t$, which is defined as the average number of secondary cases infected by an infected case with symptom onset at day $t$. Various methods to estimate $R_t$ under different scenarios were proposed in the literature [18–23].

If complete information about who infects whom is known, $R_t$ can be determined by simply counting secondary cases. However, tracing information is usually incomplete or not timely available, and thus statistical approaches are required. Willinga and Teunis [24] proposed a likelihood-based method to estimate $R_t$ from the epidemic curve and the distribution of generation intervals, which works only for the period in which all secondary cases would have been detected, thus resulting in a time lag about 19 days for COVID-19 (95th percentile of the distribution of generation intervals [1]). By accounting for yet unobserved secondary cases via Bayesian inference, Cauchemez *et al.* [25] extended the Wallinga-Teunis method to provide real-time estimates of $R_t$.

In real world, the situation may be even worse, where not only the complete tracing records, but also the full epidemic curves are unknown. In order to deal with such situation, we proposed a Monte-Carlo method to estimate the full epidemic curve by using a small number of cases with known symptom onsets, and then to estimate the reproduction number.

## Materials and methods

### Estimation of $R_t$

Distribution of generation intervals and epidemic curve are two main inputs to estimate $R_t$, where generation intervals refer to time intervals between symptom onsets of index cases and their infected cases, and the epidemic curve records the number of cases with symptom onsets at each day. According to the empirical observations [1], the distribution of generation intervals, $q(t_g)$, can be approximated by a Gamma distribution [26]:

$$q(t_g) = \frac{\beta^\alpha}{\Gamma(\alpha)} t_g^{\alpha-1} e^{-\beta t_g} \ (t_g > 0),
\tag{1}$$

where $\alpha \approx 4.866$ is the shape parameter and $\beta \approx 0.649$ is the inverse scale parameter. Given two cases $i$ and $j$ with symptom onset times being $t_i$ and $t_j$, the likelihood that case $i$ is infected by case $j$ ($t_i > t_j$) is thus

$$\rho_{ij} = \frac{q(t_i - t_j)}{\sum_{k,t_i > t_k} q(t_i - t_k)}.
\tag{2}$$

Wallinga and Teunis [24] suggested that the expected number of secondary cases infected by case $j$ can be estimated by the sum of likelihoods, as

$$R_j = \sum_{i, t_i > t_j} \rho_{ij}.$$

(3)

The effective reproduction number can thus be estimated as

$$R_t = \frac{1}{|C_t|} \sum_{j \in C_t} R_j,$$

(4)

where $C_t$ is the set of cases with symptom onsets at day $t$. Obviously, $R_t = R_j$ if $j \in C_t$ since in the Wallinga-Teunis method, cases with the same symptom onset time have the same expected number of secondary cases.

We further consider the task to calculate the effective reproduction number $R_t$ given the last known onset time $T$. Obviously, only if $T > t$, this task is possible. If $T \geq t + t_g^{\max}$ with $t_g^{\max}$ denoting the maximum generation interval, we can directly apply the Wallinga-Teunis method. However, if $t < T < t + t_g^{\max}$, we need to introduce an additional step with Bayesian inference [25]. Assuming the mean number of secondary cases infected by a case with symptom onset at day $t$ can be decomposed by two parts as

$$R_t = R_t^-(T) + R_t^+(T),$$

(5)

where $R_t^-(T)$ and $R_t^+(T)$ are the mean numbers of secondary cases with symptom onsets before or at $T$ and after $T$, respectively. The value of $R_t^-(T)$ can be directly estimated by using the Wallinga-Teunis method, and thus we can infer the effective reproduction number as

$$R_t = \frac{R_t^-(T)}{\sum_{t_g=1}^{T-t} q(t_g)}.$$

(6)

## Temporal reproduction number

In this paper, we also consider a slightly different reproduction number, called the temporal reproduction number, to include the period-dependent metric $R_{[t_1, t_2]}(t_1 \leq t_2)$ that is defined as the average number of secondary cases infected by an infected case with symptoms onset during the time period $[t_1, t_2]$ [27]. Accordingly, $R_t$ is a special case of $R_{[t_1, t_2]}$ when $t_1 = t_2 = t$. Similar to the effective reproduction number, the temporal reproduction number can be estimated as

$$R_{[t_1, t_2]} = \frac{1}{|C_{[t_1, t_2]}|} \sum_{j \in C_{[t_1, t_2]}} R_j,$$

(7)

where $C_{[t_1, t_2]}$ is the set of cases with symptom onsets in the range $[t_1, t_2]$.

## Inferring the epidemic curve

For both methods proposed by Willinga and Teunis [24] and Cauchemez *et al.* [25], the epidemic curve must be given so as to estimate the effective reproduction number or temporal reproduction number. However, we usually face an even-worse condition about data

accessibility, where not only the complete tracing records, but also the full epidemic curve is unknown. For example, the number of confirmed cases of COVID-19 for each province in mainland China is made public every day, while the symptom onset of each case is not reported by Chinese CDC. Using the collected records with both known symptom onsets and confirmed dates from scattered reports, we can obtain the empirical distribution of time intervals between symptom onsets and laboratory confirmations, say $p(t_\Delta)$. Then, we develop a Monte-Carlo method to infer the epidemic curve. Given a case $i$ confirmed at day $t^{(i)}$, sample a time interval $t_\Delta^{(i)}$ according to the distribution $p(t_\Delta)$ and set $i$'s symptom onset as $t_i = t^{(i)} - t_\Delta^{(i)}$. Specifically, the uniform stochastic model $U(0, 1)$ is used to sample time intervals between symptom onsets and laboratory confirmations. that is, we use uniform stochastic model $U(0, 1)$ to return a random number $z$ between 0 and 1, and then the time interval $t_\Delta^{(i)}$ is defined by the constrain $P(t_\Delta^{(i)} - 1) < z \le P(t_\Delta^{(i)})$, where $P(t_\Delta)$ is the cumulative distribution corresponding to $p(t_\Delta)$. Combining it with the methods mentioned above, we can estimate effective reproduction number and temporal reproduction number, and thus evaluate the efficacy of control measures.

In this paper, we implement $S = 10000$ independent runs to obtain the mean values and confidence intervals. Furthermore, we take the interval time between the symptom onsets and laboratory confirmations as the statistic variable **X**, and use K-S test [28] to estimate the marginal error $\varepsilon$, as

$$\varepsilon = D_\alpha \sigma, \tag{8}$$

$$D_\alpha = 0.888/\sqrt{S}, \tag{9}$$

$$\sigma = \sqrt{\frac{1}{S}\sum_{i=1}^{S} X_i^2 - \left(\frac{1}{S}\sum_{i=1}^{S} X_i\right)^2}, \tag{10}$$

where $S$ is the sample size (i.e., the number of independent runs), $\sigma$ is the standard deviation, $\alpha$ is the significance level, and $D_\alpha$ is the critical value. In our work, the marginal error is $\varepsilon = 0.0379$ subject to $\alpha = 0.05$ and $S = 10000$.

In summary, the proposed method can be decomposed into three parts, namely, inputs, output and processes. The inputs include the distribution of generation intervals, the symptom onsets of some cases, and the laboratory confirmations of all cases. The output of the model is the estimated effective reproduction number $R_t$. In the processes, we estimate the distribution of intervals between symptom onsets and laboratory confirmations based on the cases with known symptom onsets and laboratory confirmations and apply the Monte Carlo sampling method to estimate the symptom onsets of other cases based on their laboratory confirmations. So that, the epidemic curve of all cases can be approximately obtained. Finally, the effective reproduction number is estimated according to the epidemic curve and the distribution of generational intervals. The inputs, output and processes of the proposed method are illustrated in Fig 1.

## Results

We have collected all 76936 confirmed cases reported in official websites, which are the known ensemble for the mainland China from 11 January 2020 to 22 February 2020. The detailed quantitative information of daily number of confirmed cases is from National Health Commission of China whose URL address is http://www.nhc.gov.cn/xcs/yqtb/list_gzbd.shtml. A very small fraction (4.74%) of these confirmed cases (i.e 3650 cases) with known symptom

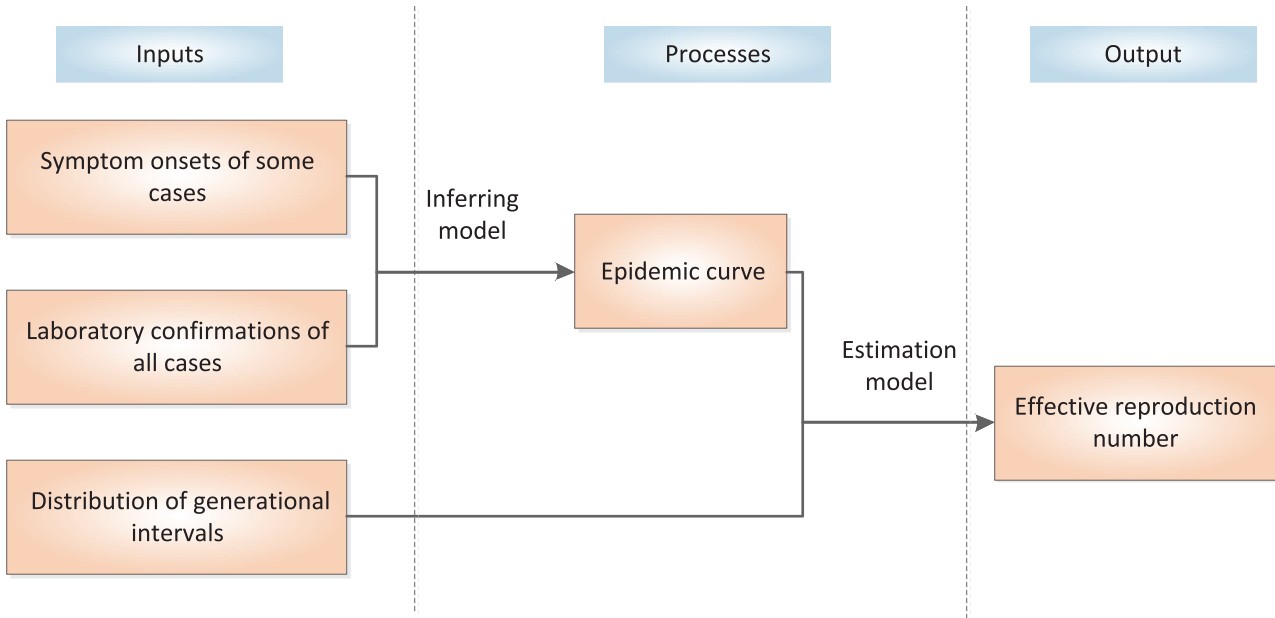

**Fig 1. The inputs, output and processes of the proposed method.**

onsets are collected from the six provinces that have reported such information. Since all provinces except Hubei applied almost the same control measures, the samples are representative. The confirmed cases for Tibet and Qinghai are only 1 and 15, so we do not analyze these two provinces.

Based on the six provinces with records of symptom onsets, we have checked that individual distributions are close to each other and can be well resembled by the synthesized distribution (see Fig 2).

Moreover, as shown in Fig 3, the synthesized distribution $p(t_\Delta)$ can be well fitted by a translational Weibull distribution [29]:

$$p(t_\Delta) = \frac{\alpha}{\beta} \left( \frac{t_\Delta + \gamma}{\beta} \right)^{\alpha-1} e^{-\left(\frac{t_\Delta + \gamma}{\beta}\right)^{\alpha}}, \tag{11}$$

where the shape parameter $\alpha \approx 1.48$, the scale parameter $\beta \approx 7.03$, and the translational parameter $\gamma = 0.10$. We introduce the translational parameter because some cases are confirmed immediately so $p(0) > 0$, while the original Weibull distribution gives $p(0) = 0$ for any shape parameter and scale parameter.

The province-level results are shown in Table 1. These results demonstrate the impressive achievement by control measures, namely $R_t$ for the majority of provinces decreased to $< 1$ within one week from the starting date of control. Even for Hubei, the epidemic was under control ($R_t < 1$) in just two weeks. In addition, within a month, the average temporal reproduction number over all provinces already decayed to 0.18, a very small value corresponding to a dying phase of the epidemic. Fig 4 reports the estimated $R_t$ for each province from 10 January 2020 to 21 February 2020 by using the present method.

Furthermore, we propose a so-called 5$\Gamma$-model with $N = 1, 000, 000$ individuals to illustrate the reliability of the present method. The spreading starts with 10 initially infected individuals, and all infected and susceptible individuals are fully mixed. In the simulation, in each time step (i.e., a day), the number of contacted individuals of each infected case is independently

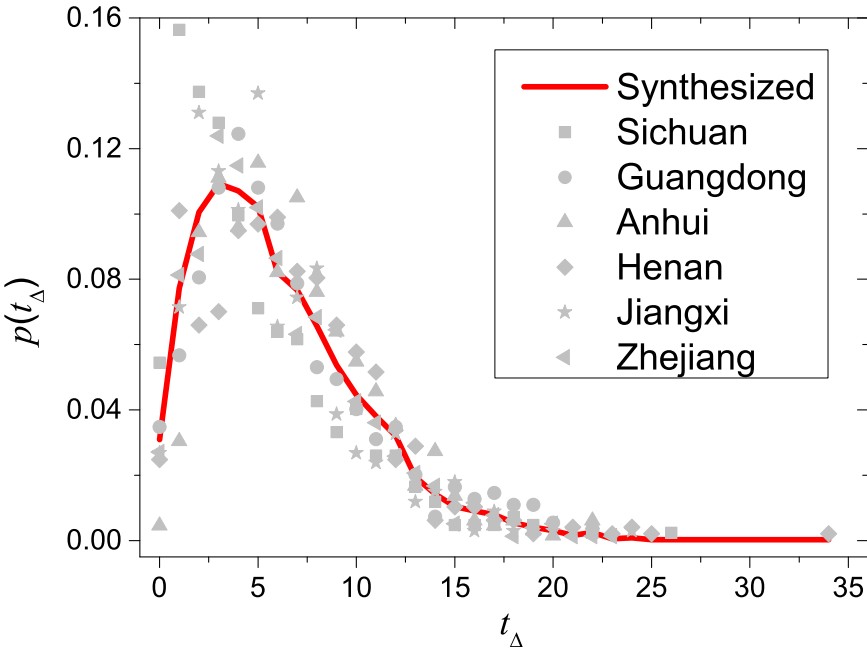

**Fig 2. Comparison between the synthesized distribution of time intervals between symptom onsets and confirmations (red solid line) and individual distributions of Sichuan, Guangdong, Anhui, Henan, Jiangxi and Zhejiang (gray data points).**

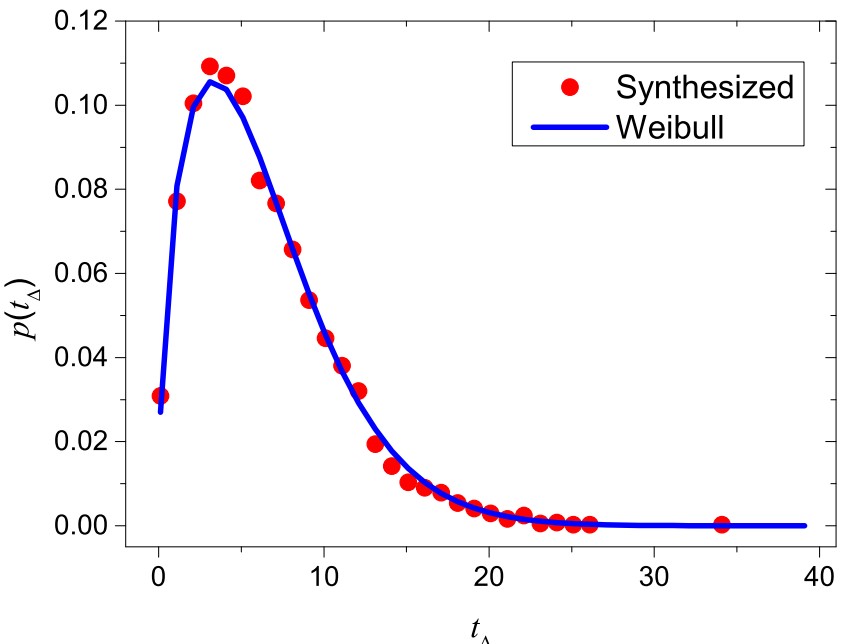

**Fig 3. Comparison between the synthesized distribution of time intervals between symptom onsets and confirmations (red circles) and the fitting curve (blue curve) that obeys the translational Weibull distribution (11).**

**Table 1. Results for all provinces in mainland China except Tibet and Qinghai, where the confirmed cases are too few to do statistics.** For each province, we show: (i) the number of cumulated confirmed cases by 22 February 2020; (2) the date $t^*$ when $R_t$ got below 1; and (iii) the temporal reproduction number during the last week [15 February 2020, 21 February 2020]. The results are averaged over 10000 independent runs.

| Province | Number of cumulated confirmed cases | Date $t^*$ when $R_t$ below 1 | Temporal reproduction number of the last week |
|---|---|---|---|
| Fujian | 298 | 2020/1/23 | 0.1365 |
| Liaoning | 121 | 2020/1/23 | 0.0053 |
| Yunnan | 174 | 2020/1/23 | 0.2039 |
| Shanghai | 335 | 2020/1/24 | 0.1967 |
| Zhejiang | 1205 | 2020/1/24 | 0.2895 |
| Chongqing | 573 | 2020/1/24 | 0.2463 |
| Beijing | 399 | 2020/1/25 | 0.2493 |
| Gansu | 91 | 2020/1/25 | 0 |
| Guangdong | 1342 | 2020/1/25 | 0.1088 |
| Guangxi | 249 | 2020/1/25 | 0.3232 |
| Hunan | 1016 | 2020/1/25 | 0.1321 |
| Shaanxi | 245 | 2020/1/25 | 0.3002 |
| Sichuan | 526 | 2020/1/25 | 0.1757 |
| Henan | 1271 | 2020/1/26 | 0.0848 |
| Nei Monggol | 75 | 2020/1/26 | 0.3176 |
| Ningxia | 71 | 2020/1/26 | 0.0146 |
| Shanxi | 132 | 2020/1/26 | 0.278 |
| Shandong | 754 | 2020/1/27 | 0.4977 |
| Anhui | 989 | 2020/1/27 | 0.082 |
| Hainan | 168 | 2020/1/27 | 0.3487 |
| Jiangsu | 631 | 2020/1/27 | 0.0901 |
| Jiangxi | 934 | 2020/1/27 | 0.0556 |
| Tianjin | 135 | 2020/1/27 | 0.4241 |
| Hebei | 311 | 2020/1/28 | 0.1736 |
| Jilin | 91 | 2020/1/28 | 0.1651 |
| Guizhou | 146 | 2020/1/29 | 0.0156 |
| Heilongjiang | 480 | 2020/1/29 | 0.1307 |
| Xinjiang | 76 | 2020/1/30 | 0.132 |
| Hubei | 64287 | 2020/2/2 | 0.0491 |

drawn from the Gamma distribution $\Gamma_1$. For each contact between an infected individual and a susceptible individual, the infected probability is independently drawn from the Gamma distribution $\Gamma_2$. The time intervals between symptom onsets and laboratory confirmations obey the Gamma distribution $\Gamma_3$. The generation intervals obey the Gamma distribution $\Gamma_4$. The time intervals between laboratory confirmations and removals from the dynamics (i.e., died, recovered, effectively isolated, etc.) obey the Gamma distribution $\Gamma_5$. The means and variances of all the five Gamma distributions are listed in Table 2.

We assume that the symptom onsets of 20% randomly selected confirmed cases are known, and the laboratory confirmations of all cases are known. The effective reproduction number $R_t$ can be directly counted by the simulation model as all transmission chains are known. We compare the accuracy of our method and that of the Wallinga-Teunis method, with simulation results being the benchmark. As shown in Fig 5, the effective reproduction numbers estimated by our method are very close to the benchmark values and remarkably more accurate than those obtained by the Wallinga-Teunis method. We have also checked that our estimations work well subject to other reasonable settings of distributions and parameters.

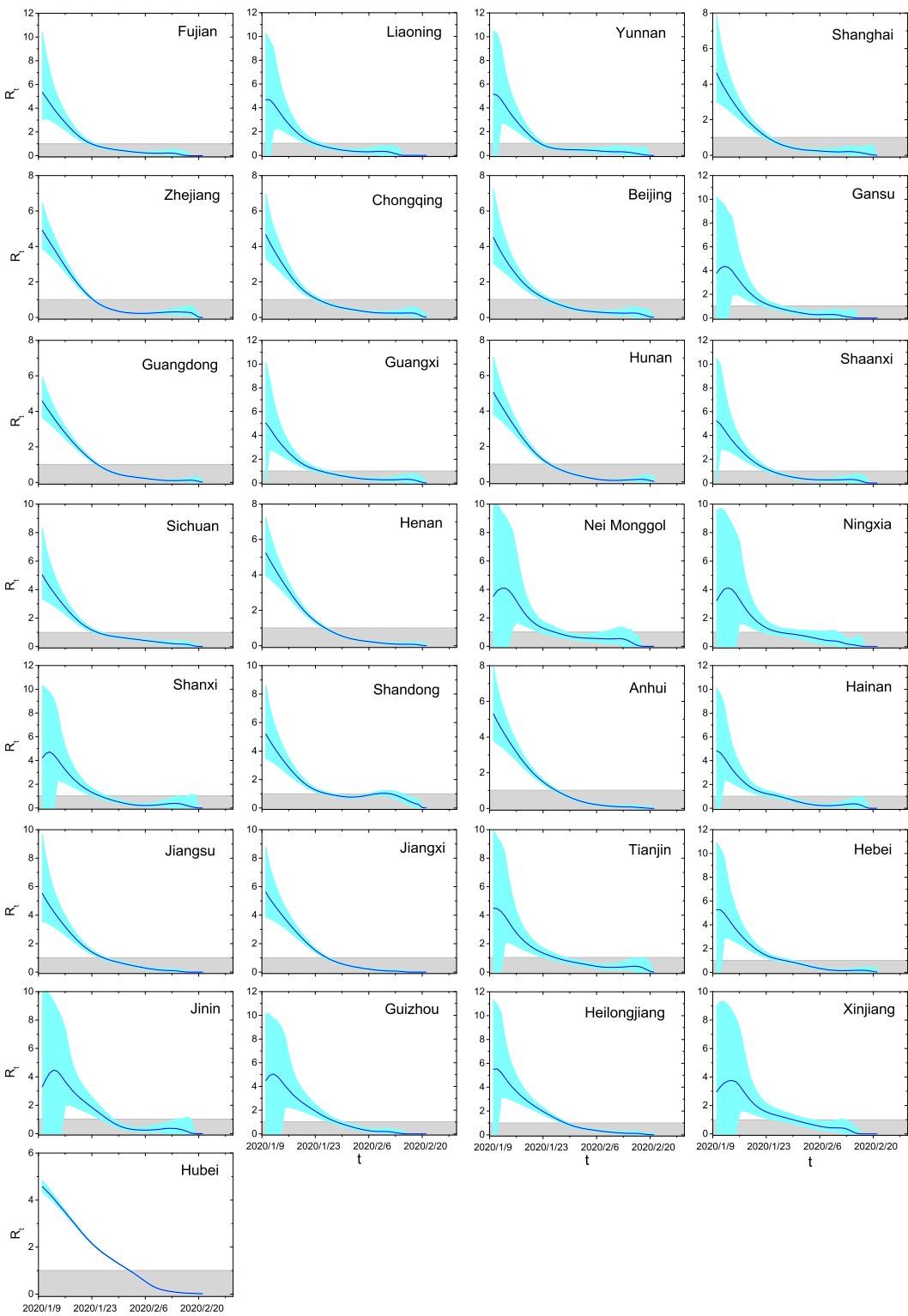

**Fig 4. Effective reproduction numbers for all provinces in mainland China from 10 January 2020 to 21 February 2020.**
The results are averaged over 10000 independent runs, and the cyan areas denote the 95% confidence intervals. In each run, the Monte-Carlo sampling method is applied to infer the symptom onsets. The gray shadows emphasize the situations where the epidemic is under control ($R_t < 1$).

**Table 2. The means and variances of the five Gamma distributions used in the simulation model.**

| Distribution | Mean | Variance |
|:---:|:---:|:---:|
| $\Gamma_1$ | 15 | 10 |
| $\Gamma_2$ | 0.009 | $1.8 \times 10^{-6}$ |
| $\Gamma_3$ | 5 | 2 |
| $\Gamma_4$ | 7.5 | 3.4 |
| $\Gamma_5$ | 20 | 8 |

## Discussion

A Monte-Carlo method is proposed to infer the epidemic curve, and then estimate the temporal reproduction number. Our results suggest that Chinese control measures are likely to be effective and efficient, with daily number of confirmed cases quickly decreasing after a short expansion lasting about two weeks from 21 January 2020. By introducing a Monte-Carlo method to estimate the symptom onsets of confirmed cases based on a small number of cases with known symptom onsets, our method can utilize the information of all cases to calculate the effective reproduction number. In comparison, the Wallinga-Teunis method can only make use of the cases with both known symptom onsets and laboratory confirmations. As shown in Fig 5, our method produces obviously more accurate results than the Wallinga-Teunis method. One underlying assumption in our method is that the small number of samples are representative of all cases. This is a reasonable assumption for mainland China since control measures in different provinces are very much the same, all executing directives from the central government. However, in general, if the samples and the inferred cases are in different spreading stages or different areas, the reliability of the present method has to be carefully

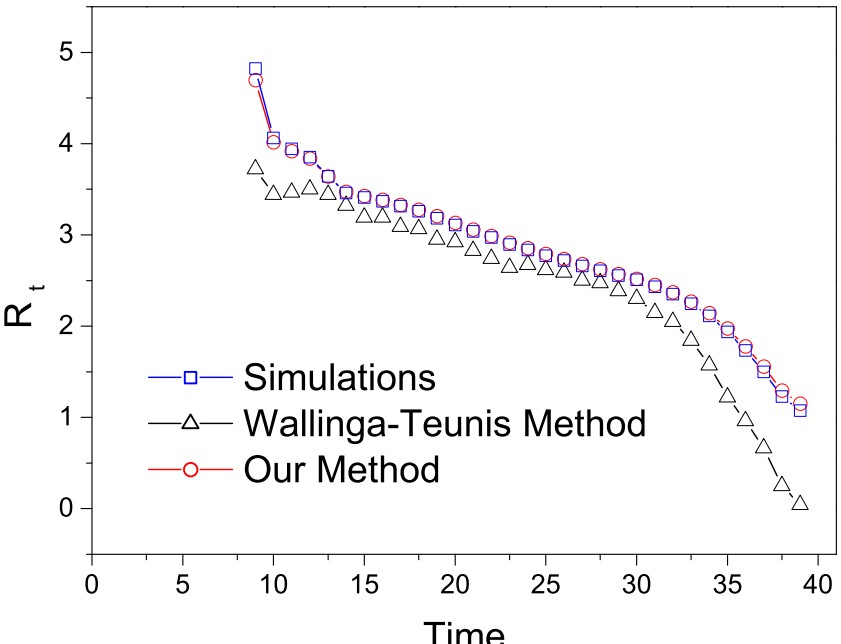

**Fig 5. The comparison of effective reproduction numbers directly counted based on the simulation results (blue squares) and estimated by the Wallinga-Teunis method (black triangles) and our method (red circles).** The results obtained by the Wallinga-Teunis method and our method are both averaged over 10000 independent runs.

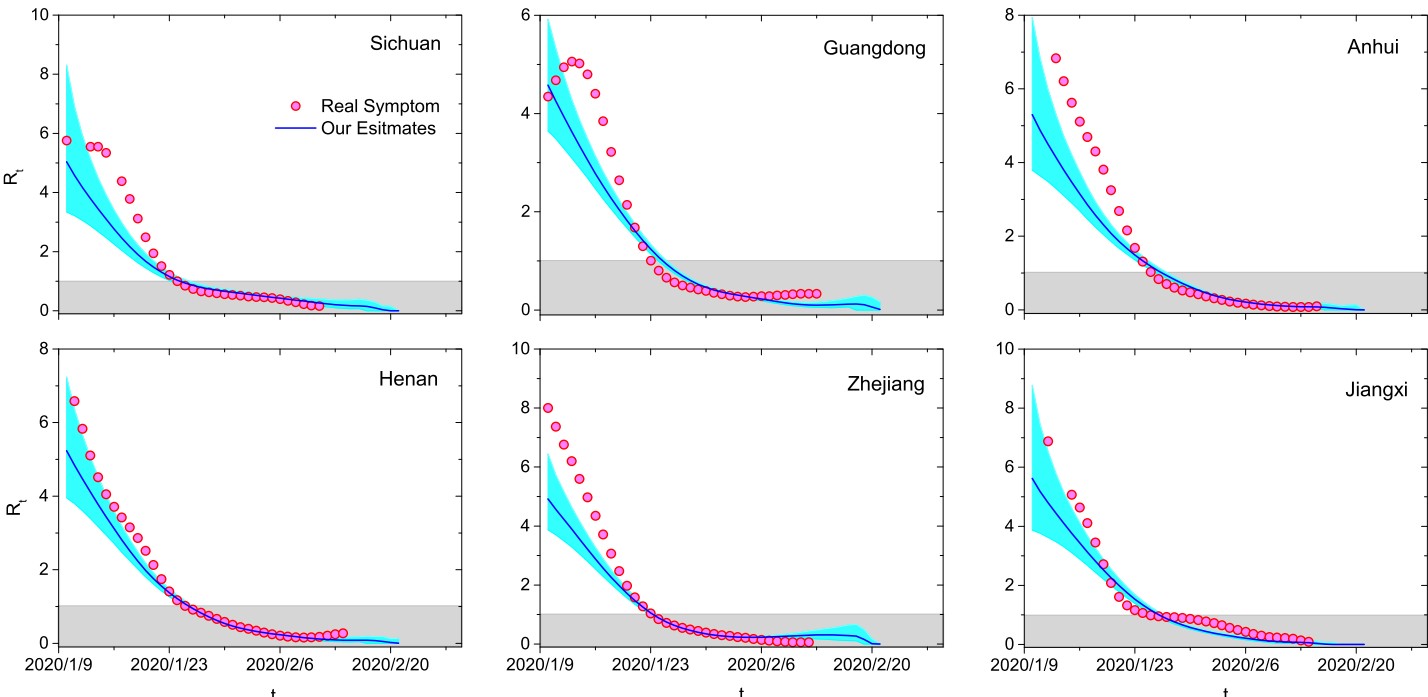

**Fig 6. Comparison between the estimates of effective reproduction numbers by the true and inferred records of symptom onsets.** The solid blue curves and cyan areas respectively denote the average values and 95% confidence intervals obtained by 10000 independent runs according to the inferred data. The red circles represent the results obtained by the true records. The gray shadows emphasize the situations where the epidemic is under control ($R_t < 1$). The six plots are results for Sichuan, Guangdong, Anhui, Henan, Jiangxi and Zhejiang.

checked before any applications. For example, in US, cases in a few states cannot represent the whole country since different states may adopt different controlling strategies and launch different control measures.

The distribution $p(t_\Delta)$ is not stable, usually with smaller and smaller mean and standard deviation in the progress of an epidemic [18]. Fig 6 compares the estimates of effective reproduction numbers by the true and inferred records of symptom onsets for the six provinces with known symptom onsets. At the very beginning, the estimates from inferred data are smaller than the ones from true records, but they are getting closer and closer and show almost the same $t^*$ in the later stage. Indeed, we still overestimate the reproduction number in the early stage, because a large fraction of cases (except Hubei) are importations [18, 30]. Fortunately, the present method shows accordance with the one accounting for importations. For example, $R_t$ of the three example provinces (Guangdong, Hunan and Shandong) approach 1 at 23 January 2020, 26 January 2020 and 30 January 2020 by the method in [30] and at 25 January 2020, 25 January 2020 and 27 January 2020 by the present method. In a word, this method can be further improved by considering importations [18, 30] and using Markov-Chain Monte-Carlo algorithm based on independent transmission assumption [31–33].

Government-led actions likely played a role in the reduction of new COVID-19 cases. In order to block transmission and reduce public health hazards, the "five early" measures, namely "early detection, early report, early investigation, early isolation and early treatment", are implemented. *Early detection*.—Rapid detection and diagnosis to promote the timely and effective management of confirmed and suspected cases. *Early report*.—Immediate report to the disease control department about confirmed and suspected cases to start investigation and treatment as soon as possible. *Early investigation*.—Quick epidemiological investigation on the

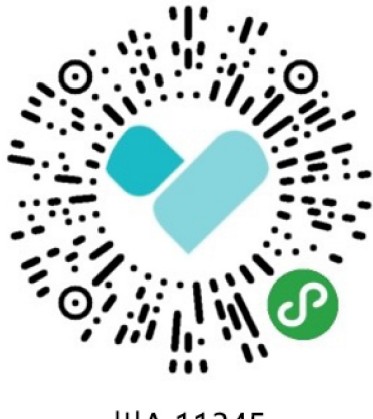

JⅡA 11345

**Fig 7. Illustration of an example of the QR codes to trace the epidemic in mainland China.** This is the one posted in a public bus in Chengdu City. In the bottom, a Chinese character followed by A11345 is the plate number of this bus, and the character is the abbreviation of Sichuan Province.

exposure and detailed contacts of confirmed and suspected cases. Through such investigation, we can find out the transmission chain of each case, so as to comprehensively manage all possible infected individuals related to each case. *Early isolation.*—All confirmed and suspected cases, as well as their close contacts will be isolated as soon as possible. *Early treatment.*— Quick providing of proper treatment (symptomatic treatment, supportive treatment, antiviral treatment via traditional Chinese medicine, etc.) to prevent the development of symptom. To efficiently and effectively implement the "five early" measures, some advanced information techniques are employed to trace the epidemic spreading. For example, in many cities, the QR codes [34, 35] (similar to these used for online payments) are posted in public transport means (buses, subway stations, taxies, etc.), places with possible crowds (supermarkets, bazaars, restaurants, office buildings, etc.) and places worth particular attention (drugstores). People are asked to scan the codes before entering, so the administrators can get the corresponding check-in records with identifications (mobile phone ID). Therefore, if a person is laboratory confirmed or identified as a suspected case, the administrators will know immediately and exactly the persons who have possible contacts with this case by simply searching the check-in records. This operation is completely automatic with private information being protected if an individual is not laboratory confirmed, suspected or having close contacts with the above two kinds of people (even one is confirmed, her/his personal information is only used in fighting the disease). Fig 7 illustrates an example of the QR codes, which was posted in a bus in Chengdu City of Sichuan Province, and people are required to scan the code before getting on the bus. Therefore, if a confirmed or suspected case has taken this bus, we can immediately find out people who have also taken this bus in the same time period. This is in our opinion a simple but perfect tool in the epidemiological perspective to efficiently and effectively block the spread through communities.

## Supporting information

**S1 Dataset.**
(RAR)

## Acknowledgments

We thank Yan Wang, Wei Bai and Min Wang for data collection, Qin Gu for developing the check-in system via scanning the QR codes and sharing some representative QR codes with us, and Quan-Hui Liu for helpful discussion.

## Author Contributions

**Formal analysis:** Tao Zhou.

**Investigation:** Duanbing Chen.

**Methodology:** Duanbing Chen, Tao Zhou.

**Software:** Duanbing Chen.

**Writing – original draft:** Duanbing Chen, Tao Zhou.

**Writing – review & editing:** Duanbing Chen, Tao Zhou.

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
