## [Decision Letter · Decision Letter 0]

3 Jul 2020

PONE-D-20-11733

Chinese Control Efficacy on COVID-19

PLOS ONE

Dear Dr. Zhou,

Thank you very much for submitting your manuscript "Chinese Control Efficacy on COVID-19" (#PONE-D-20-11733) for review by PLOS ONE. As with all papers submitted to the journal, your manuscript was fully evaluated by academic editor (myself) and by independent peer reviewers. The reviewers appreciated the attention to an important health topic, but they raised substantial concerns about the paper that must be addressed before this manuscript can be accurately assessed for meeting the PLOS ONE criteria. Therefore, if you feel these issues can be adequately addressed, we invite you to submit a revised version of the manuscript that addresses the points raised during the review process. We can’t, of course, promise publication at that time.

We look forward to receiving your revised manuscript.

Kind regards,

Abdallah M. Samy, PhD

Academic Editor

PLOS ONE

Journal Requirements:

2. In the Methods, please clarify how information about the daily number of confirmed cases for all provinces in mainland China from 11 January 2020 to 22 February 2020 was collected, including the source of the data. Please ensure that sufficient information is provided so that other researchers could potentially replicate these analyses.

3.We note that you have indicated that data from this study are available upon request. PLOS only allows data to be available upon request if there are legal or ethical restrictions on sharing data publicly. For more information on unacceptable data access restrictions, please see http://journals.plos.org/plosone/s/data-availability#loc-unacceptable-data-access-restrictions.

**Reviewers' comments:**

Reviewer's Responses to Questions

**Comments to the Author**

1. Is the manuscript technically sound, and do the data support the conclusions?

Reviewer #1: Partly

2. Has the statistical analysis been performed appropriately and rigorously?

Reviewer #1: Yes

3. Have the authors made all data underlying the findings in their manuscript fully available?

Reviewer #1: Yes

4. Is the manuscript presented in an intelligible fashion and written in standard English?

Reviewer #1: No

5. Review Comments to the Author

Reviewer #1: General comments –

• The manuscript needs editing for spell check, English and grammar.

• The control methods deployed in China should also be described considering that these were effective to get the R0 to <1 within a week. It may help other countries to streamline their strategies

Specific comments –

Sample size –

The authors have not described how the sample size was estimated? Was the study adequately powered to predict the outcomes? It is recommended that a post-hoc power analysis be undertaken to assess if study is also adequately powered to meet the study objectives.

Sampling strategy –

How or on what criteria, the sample was selected should be described? Was it representative of the other COVID-19 patients in terms of profile and severity?

Methods –

The authors have not specified what type of distribution their data followed (normal/uniform/discrete/triangular/Beta-PERT distribution), this will determine

how to output a random variable that follows a certain distribution. The authors should specify this and accordingly justify the method used. Did they use any of the data transformation methods? If so this should be specified

Results –

• As Monte Carlo method is a probabilistic method with randomness playing a role in predicting future outcomes, there will always be a margin of error related to the results. The authors should specify the margin of error and confidence probability of valid findings.

• What was the accuracy of this proposed new method to the existing methods for simulation to calculate R0 that the authors have described.

• Kindly describe how exactly can/must we define the inputs and model the underlying processes to use this proposed new method?

• It is recommended that tallying of Simulation results be done to establish reliability

Discussion

• Is the Monte Carlo method that uses a stochastic model to your data? should be discussed

• Discuss the accuracy of your proposed method study vis-à-vis the accuracy of other established methods.

• Strengths and Limitations of the study should be discussed

• Study is conducted in a small sub-set of Chinese population, limitations related to external generalizability should be discussed

Ethical considerations/obligations

The manuscript is silent about the ethical considerations/obligations.

• Was an approval taken from any ethics committee?

6. PLOS authors have the option to publish the peer review history of their article (what does this mean?). If published, this will include your full peer review and any attached files.

Reviewer #1: No

---

## [Author Response · Author response to Decision Letter 0]

27 Sep 2020

Thank you very much for processing our manuscript entitled “Chinese Control Efficacy on COVID-19”, and thanks for all the valuable comments and suggestions, which provide the excellent guidance to improve our manuscript. Accordingly, we have largely revised the manuscript. Enclosed please find a detailed response to the referee report. For the sake of convenience, the main modifications are marked in red in the revised manuscript. We believe that the revised manuscript can meet the standard of PLoS ONE.

---

## [Editor Report · Decision Letter 1]

12 Jan 2021

PONE-D-20-11733R1

Chinese Control Efficacy on COVID-19

PLOS ONE

Dear Dr. Zhou,

Thank you very much for submitting your manuscript "Chinese Control Efficacy on COVID-19" (#PONE-D-20-11733R1) for review by PLOS ONE. As with all papers submitted to the journal, your manuscript was fully evaluated by academic editor (myself) and by independent peer reviewers. The reviewers appreciated the attention to an important health topic, but they raised substantial concerns about the paper that must be addressed before this manuscript can be accurately assessed for meeting the PLOS ONE criteria. Therefore, if you feel these issues can be adequately addressed, we invite you to submit a revised version of the manuscript that addresses the points raised during the review process. We can’t, of course, promise publication at that time.

We look forward to receiving your revised manuscript.

Kind regards,

Abdallah M. Samy, PhD

Academic Editor

PLOS ONE

**Additional Editor Comments:**

Please address carefully all our comments below. Thanks!

1. We note that the authors state in their abstract "The province-level analysis indicates that Chinese control measures on COVID-19 are very effective and efficient, that is, the effective reproduction numbers of the majority of provinces in mainland China got down to < 1 just by one week from the setting of control measures, and the temporal reproduction number of the week [15 Feb, 21 Feb] is only about 0.18" and also state in their discussion "The results indicate that Chinese control measures have achieved remarkable success..." and "The huge success of Chinese control measures on COVID-19 resulted from the ambitious and aggressive government-led actions." However, their study does not directly test whether specific control measures caused the reduction of R and new cases, and thus, we do not feel that these statements are supported by the rest of the study. To meet our publication criteria that conclusions are supported by the data

(https://journals.plos.org/plosone/s/criteria-for-publication#loc-4) we recommend that authors change these sentences to something such as:

Abstract:

"The province-level analysis indicates that the effective reproduction numbers of the majority of provinces in mainland China got down to < 1 just by one week from the setting of control measures, and the temporal reproduction number of the week [15 Feb, 21 Feb] is only about 0.18. It is therefore likely that Chinese control measures on COVID-19 were effective and efficient, though more research needs to be performed."

Discussion:

"Our results suggest that Chinese control measures have been effective..." and

"Government-led actions likely played a role in the reduction of new COVID-19 cases."

2. We also note that PLOS’ guidelines state that the title should be "specific, descriptive, concise, and comprehensible to readers outside the field" (https://journals.plos.org/plosone/s/submission-guidelines#loc-title). In this case, we feel that the title is vague and does not describe the methods or aims of the study. We suggest that the title include a reference to the methodology or component to be measured (i.e., temporal reproduction number), the aim of the study (i.e, evaluating the effect of implementing COVID-19 control measures on reproduction number"), and the locale that was studied (i.e., China). For example, a title such as "COVID-19 control measure implementation in China: estimating the effect on temporal reproduction number" would be appropriate.

---

## [Author Response · Author response to Decision Letter 1]

21 Jan 2021

We have revised the manuscript according to the editor's suggestions. For the sake of convenience, the main modifications are marked in red in the revised manuscript.

---

## [Editor Report · Decision Letter 2]

26 Jan 2021

Evaluating the effect of Chinese control measures on COVID-19 via temporal reproduction number estimation

PONE-D-20-11733R2

Dear Dr. Zhou,

We’re pleased to inform you that your manuscript, "Evaluating the effect of Chinese control measures on COVID-19 via temporal reproduction number estimation" (PONE-D-20-11733R2), has been judged scientifically suitable for publication and will be formally accepted for publication once it meets all outstanding technical requirements.

Kind regards,

Abdallah M. Samy, PhD

Academic Editor

PLOS ONE

---

## [Editor Report · Acceptance letter]

1 Feb 2021

PONE-D-20-11733R2 

Evaluating the effect of Chinese control measures on COVID-19 via temporal reproduction number estimation 

Dear Dr. Zhou:

I'm pleased to inform you that your manuscript has been deemed suitable for publication in PLOS ONE. Congratulations! Your manuscript is now with our production department. 

Kind regards, 

on behalf of

Dr. Abdallah M. Samy 

Academic Editor

PLOS ONE